# The Benefits of Using Saccharose for Photocatalytic Water Disinfection

**DOI:** 10.3390/ijms23094719

**Published:** 2022-04-25

**Authors:** Paulina Rokicka-Konieczna, Agata Markowska-Szczupak, Ewelina Kusiak-Nejman, Antoni W. Morawski

**Affiliations:** 1Department of Inorganic Chemical Technology and Environment Engineering, Faculty of Chemical Technology and Engineering, West Pomeranian University of Technology in Szczecin, Pułaskiego 10, 70-322 Szczecin, Poland; ewelina.kusiak@zut.edu.pl (E.K.-N.); antoni.morawski@zut.edu.pl (A.W.M.); 2Department of Chemical and Process Engineering, Faculty of Chemical Technology and Engineering, West Pomeranian University of Technology in Szczecin, Al. Piastów Ave. 42, 71-065 Szczecin, Poland; agata.markowska@zut.edu.pl

**Keywords:** photocatalysis, sucrose-modified titanium dioxide, disinfection

## Abstract

In this work, the characteristics of saccharose (sucrose)-modified TiO_2_ (C/TiO_2_) photocatalysts obtained using a hydrothermal method at low temperature (100 °C) are presented. The influence of C/TiO_2_ on survivability and enzyme activity (catalase and superoxide dismutase) of Gram-negative bacteria *Escherichia coli* (ATCC 29425) and Gram-positive bacteria *Staphylococcus epidermidis* (ATCC 49461) under UV-A and artificial solar light (ASL) were examined. The obtained TiO_2_-1%-S-100 photocatalysts were capable of total *E. coli* and *S. epidermidis* inactivation under ASL irradiation in less than 1 h. In addition, the impacts of sugars on the photocatalytic activity and disinfection performance are discussed.

## 1. Introduction

The World Health Organization (WHO) defined the need to search for effective water disinfections methods [1]. Solar water disinfection (SODIS) is an economically and environmentally friendly option for water treatment. The most abundant renewable energy source, sunlight, is used in direct solar disinfection. However, this technology is particularly worthwhile in sunny regions of the world. Therefore, many initiatives are being taken to improve SODIS and combine it with other water treatment methods to maximize access to drinking water, particularly in developing countries [2,3].

The optimization of solar energy exploitation in water disinfection is possible by combining it with advanced oxidation processes (AOPs) [4,5]. For many years, TiO_2_ photocatalysis has been considered one of the most profitable environmental cleanup technologies [6,7]. Unfortunately, titanium dioxide application is limited in the visible light region, mainly due to its relatively wide band gap (3.2 eV) [8]. Consequently, the development of new TiO_2_ photocatalysts active under solar irradiation is a field of particular scientific interest [9,10]. A literature review revealed that non-metal modifications have been presented as promising and inexpensive choices for improving TiO_2_ activity. Many authors reported that incorporating carbon into TiO_2_ materials led to obtaining photocatalysts with enhanced efficiency and activity under UV and visible light [11,12,13,14,15]. Moreover, it was reported that carbon-doped titanium dioxide is more active under visible light than nitrogen- or sulfur-doped options [12,14]. In our previous work [16,17], the modification of titanium dioxide using the simple sugars D-glucose and D-fructose resulted in the preparation of C/TiO_2_ photocatalysts, which exhibited satisfactory antibacterial properties against both Gram-positive and Gram-negative bacteria under artificial solar light. It was also summarized that the low cost of monosaccharides makes them attractive dopants for modifications of titania at low temperatures. It is well-known that both fructose and glucose are manufactured at an industrial scale for food production using polysaccharides that are abundant in nature (mainly starch) [18]. The main goal of the present study is to contribute to the understanding of the influence of sugar (saccharides, carbohydrates) types on the antibacterial properties of titania. A natural choice for this study is a sucrose non-reducing disaccharide of glucose that yields two monosaccharides: glucose and fructose. To the best of our knowledge, there is no information regarding the use of double sugars to fabricate a novel titania photocatalyst for solar water disinfection.

## 2. Results and Discussion

### 2.1. Characterization of Saccharose-Modified TiO_2_

In order to obtain a novel photocatalyst active under sunlight, titanium dioxide taken directly from the production line of TiO_2_ was treated with solutions of sucrose (Firma Chempur^®^, Piekary Śląskie, Poland) at various concentrations (1, 5 and 10 wt.%) and with annealing at a temperature of 100 °C. The surface properties of examined photocatalysts were investigated utilizing FT-IR/DR spectroscopy. Figure 1 presents FT-IR/DR spectra of the starting TiO_2_-100, saccharose-modified TiO_2_ photocatalysts, pristine saccharose and KRONOClean 7000 as reference materials.

The presented spectra exhibit bands typical for TiO_2_-based nanoparticles and saccharose. A broad band observed at 3700–2800 cm^−1^ corresponds to O–H stretching vibration [19]. The narrow band at 1621 cm^−1^ was attributed to the molecular water bending mode [20]. In turn, the strong peak at 960 cm^−1^ corresponds to the self-absorption of titanium (Ti^4+)^ [21]. It should be noted that the bands located at 3700–2800 cm^−1^ and 1621 cm^−1^ in saccharose-modified TiO_2_ were broader and more intensive than those of unmodified TiO_2_-100. This indicates that the hydroxyl groups from saccharose could bond to the TiO_2_ surface, according to data for the TiO_2_-glucose surface complex presented by Kim et al. [22]. The obtained result was also in agreement with data obtained by Dong et al. [23]. The author modified TiO_2_ in the hydrothermal process at 160 °C with glucose as a carbon source. Additionally, it is worth mentioning that a greater amount of hydroxyl groups on a photocatalyst’s surfaces will enhance the photocatalytic activity [24]. It can be observed that saccharose-modified photocatalysts had new characteristic bands between 1500 and 400 cm^−1^. The band at 1418 cm^−1^ is a combination of O–H bending of the C–OH group and C–H bending in the carbohydrate. The band at 1110 cm^−1^ corresponds to the stretching of the C–O band of the C–O–C linkage [24]. According to Tul’chinsky et al. [25], the band at 869 cm^−1^ was assigned to the vibration of the furanose ring. A stretched C–O mode characteristic for saccharose was noted at 1053 cm^−1^. In turn, bands at 695 cm^−1^ and 417 cm^−1^ were associated with the C–C stretching modes [26]. A peak at 405 cm^−1^ corresponded to the glucopyranose ring deformations [26]. Sucrose is a disaccharide, and it contains two sugar rings. It is made up of one molecule of glucose (aldohexose) and one molecule of fructose (ketohexose) joined together by glycosidic linkages [27]. In the spectrum obtained for KRONOClean 7000, a band at 1580 cm^−1^ was attributed to the asymmetric and symmetric stretching vibrations of arylcarboxylate groups. The peak at 1330 cm^−1^ was assigned to the C–O stretching vibrations [28]. It is clear that KRONOClean 7000 is a carbon-modified commercial anatase displaying pronounced photoactivity in visible light. The XRD patterns of tested C/TiO_2_ photocatalysts are presented in Figure 2.

As shown above, after carbon modification, samples present similar patterns compared to unmodified TiO_2_-100. All tested photocatalysts consist mainly of an anatase phase of approximately 98%. The characteristic peaks for anatase structures observed at 2θ values of 25.3, 37.6, 47.8, 53.7, 62.6, 70.2 and 75.0° were attributed to (101), (004), (200), (105), (204), (116) and (215) crystal planes (JCPDS 01-070-7348). Additionally, two small peaks located at 27.1° and 36.0° were found, which corresponded to the rutile crystal planes (110) and (101) (JCPDS 01-076-0318). The presence of rutile in TiO_2_ photocatalysts amounted to approximately 2%, which resulted from rutile nuclei added in the production process [17]. Commercial KRONOClean 7000 consisted only of the anatase phase consistent with the manufacturer’s information. The average crystallite sizes of the obtained photocatalysts were determined using Scherrer’s equation [29] and are listed in Table 1.

All carbon-modified samples showed similar anatase crystallite sizes of about 11.6–11.7 nm. Commercial KRONOClean 7000 contains an anatase phase with a mean crystallite size of about 11 nm [30]. It should be mentioned that data presented by the supplier report that the mean crystallite size of anatase in KRONOClean 7000 amounts to approximately 15 nm [31]. The saccharose modification conducted at 100 °C did not significantly influence the anatase crystallite size. However, as with the previous data obtained for modifications of titania with fructose and glucose, the smallest crystallites characterized the photocatalyst modified by a 10% monosaccharide solutions [16,17].

The N_2_ adsorption–desorption analysis was performed to investigate the surface areas and pore volumes of the tested photocatalysts. The isotherms are shown in Figure 3.

According to the IUPAC classification, all isotherms were identified as typical type IV. TiO_2_-100 and C/TiO_2_ exhibited type H4 hysteresis loops. For commercial KRONOClean 7000, a type H3 hysteresis loop was observed [32]. Synthesized samples were mesoporous materials with small numbers of macro- and micropores. Table 2 presents the S_BET_ area, total (V_total_), micropore (V_micro_) and mesopore (V_meso_) volumes of samples.

The S_BET_ for saccharose-modified TiO_2_ was in the range of 217–264 m^2^/g. Photocatalysts after modification were characterized by smaller surface areas and pore sizes compared to unmodified TiO_2_-100. Based on the presented results, these changes were related to increasing saccharose concentration (from 1 to 10 wt.% in the solution used for modification). Similar observations with the application of glucose and fructose monosaccharides as carbon sources were presented in previous studies [16,17]. The decrease in the S_BET_ area with increasing carbon precursor content could be associated with the accumulation of carbon on TiO_2_ surfaces [33]. The lowest S_BET_ was found for samples modified by a 10% saccharose solution (TiO_2_-S-10%-100). This effect has also been associated with the uneven covering of crystallites by a thin layer of dissolved saccharide.

The carbon content in carbon-modified photocatalysts was confirmed by elemental analysis of carbon and amounted from 0.53 to 4.40 wt.%. Commercial KRONOClean 7000 contained 0.96 wt.% of carbon. As expected, the percentage of carbon increased with increasing saccharose concentration in the solution (Table 2). Similar results were obtained for glucose and fructose modifications of titania [16,17]. However, a rise in annealing temperature above the caramelization temperature (150 °C for fructose, 160 °C for glucose and 186 °C for sucrose) can cause thermal decomposition of sugars and a decreased carbon content [16,34]. It is thought that a similar effect is possible under high pressure in an autoclave used for titania photocatalyst production via the hydrothermal method. As reported by Woo et al., organic acids such as formic acid, lactic acid and levulinic acid, as well as 5-hydroxymethylfurfural (HMF), increased with increasing heating temperatures (110 to 150 °C) and times (1 to 5 h) [35].

The UV–Vis diffuse reflectance spectra of TiO_2_-100, saccharose-modified TiO_2_ and KRONOClean 7000 are presented in Figure 4.

As shown in Figure 4, the TiO_2_-100 presents a typical high absorption in the UV region and no absorption in the visible region. In turn, carbon modification leads to an increase in light absorption in the visible region. Moreover, the intensity of visible light absorption increases with carbon content. This was related to the color change of photocatalysts after the saccharose modification. TiO_2_-100 and TiO_2_-S-1%-100 stayed white, whereas photocatalysts obtained by modification with 5 and 10 wt.% of saccharose were beige and light-brown. The most substantial absorption in the visible light region was noted for the TiO_2_-S-10%-100, which contained the highest amount of carbon (4.40 wt%). It should be noted that commercial titania KRONOClean 7000 also absorbed visible light, but the character of the absorption spectrum was different from saccharose-modified nanomaterials. This issue has been discussed in previous work [33]. The values of Eg for the analyzed samples were between 2.92 and 3.25 eV (Table 2). It was observed that these values slightly decreased with increasing carbon on the sample surface. The lowest band gap value (2.92 eV) was observed in the TiO_2_-S-10%-100 photocatalyst. Due to only 5% of UVA radiation (321–400 nm) reaching the earth’s surface, the high-visible light region is essential for developing disinfection using solar catalytic treatment.

To examine photocatalytic activity measured as the reactive oxidative species (ROS) (mainly corresponding to hydroxyl radicals ·OH) produced on the surfaces of sucrose-modified photocatalysts under ASL and UV irradiation, the fluorescence spectra of the formed 2-hydroxyterephthalic acid were measured [36,37]. The results are presented in Figure 5.

It was found that saccharose-modified photocatalysts generated more hydroxyl radicals than unmodified TiO_2_-100 and commercial KRONOClean 7000 under UV-A or ASL. Moreover, the faster creation of ·OH radicals from the surface was observed for TiO_2_-S-1%-100. After 90 min, the ROS submitted from TiO_2_-S-1%-100 under UV-A and ASL were 3.0 and 2.0 times higher, respectively, than for TiO_2_-100 (Figure 5). Interestingly, the generation of hydroxyl radicals under both types of irradiation was higher for photocatalysts obtained via sucrose modification than using glucose or fructose under the same conditions (solutions of the same sugar concentration). This was probably caused by the varied solubility of sugars in aqueous solutions at varying temperatures. As presented by Crestani et al., glucose solubility is higher than that of sucrose at temperatures of up to 50 °C, while fructose is more soluble than sucrose in the range from 30 to 100 °C [38]. It follows that photocatalysts obtained at 100 °C are composed of fine titania nanoparticles covered with a thin layer of adsorbed sugars, and their amounts are proportional to the concentration of the used solution at a given temperature.

### 2.2. Antimicrobial Activity of Saccharose-Modified TiO_2_

The presence of ·OH radicals gradually destroys the bacterial cell walls and causes oxidative damage to their cells structures, proteins (enzymes), and DNA, leading to death [39]. In order to examine the possible differences in antimicrobial actions of reactive oxygen species generated on surface-saccharose-modified titania towards Gram-negative and Gram-positive groups of bacteria, *Escherichia coli* and *Staphylococcus epidermidis* were selected as the model organisms. No significant changes in the number of bacteria were detected while these were kept in contact with the sucrose-modified titania in dark conditions for 90 min (Figure 6). Furthermore, this indicated no toxicity of the sucrose-modified TiO_2_ to *Escherichia coli* and *Staphylococcus epidermidis*.

The photolysis of bacterial cells under UV-A and ASL irradiation alone was not observed (results for the saline solution are presented in Figure 7 and Figure 8).

The bacteria inactivation was observed in experiments conducted with commercial KRONOClean 7000, starting TiO_2_ and sucrose-modified TiO_2_ under UV-A and ASL irradiation. The strongest antibacterial activity is presented for the photocatalyst obtained via modification with 1 wt.% of saccharose solution (TiO_2_-S-1%-100). Total *E. coli* inactivation was achieved after 55 min and 65 min with photocatalytic processes under UV-A and ASL irradiation, respectively (Figure 7). Gram-positive *Staphylococcus epidermidis* was more invulnerable to photocatalytic disinfection. Total bacteria inactivation was obtained in the same conditions after 65 under UV-A and after 80 min under ASL. TiO_2_-S-1%-100 photocatalysts presented better antibacterial properties than starting TiO_2_ in all experiments. It is worth mentioning that commercial KRONOClean 7000 caused *E. coli* and *S. epidermidis* inactivation in 80 and 85 min under UV-A and 90 minutes under ASL irradiation (Figure 7 and Figure 8).

The antibacterial properties of C/TiO_2_ can be attributed to various features such as the large surface area, high anatase phase content (approximately 98%) and small crystallite size. Zimbone et al. [40] observed that photocatalysts containing mainly anatase phases with particles around or smaller than about 11 nm are preferred. In this study, sucrose-modified TiO_2_ nanomaterials possess such properties (Table 1). However, it has been demonstrated that the antimicrobial activity of photocatalysts mainly depends on the carbon content. The strongest antibacterial activity against both species of bacteria was observed for TiO_2_-S-1%-100, containing the lowest amount of carbon on the surface (0.53 wt.%). The obtained results were in agreement with previous studies concerning the antibacterial properties of C/TiO_2_ photocatalysts modified with alcohols [41], monosaccharide–glucose [16,42] and fructose [17]. Thus, it was confirmed that solutions of monosaccharides or disaccharides at low concentrations are good carbon sources for synthesizing C/TiO_2_ photocatalysts via hydrothermal reaction at low temperatures (from 100 °C to 200 °C). The reduced photocatalytic activity caused by excessively high carbon content on the photocatalyst’s surface was also confirmed by Wanag et al. [43] and Cui et al. [44]. The authors reported that excess carbon on C/TiO_2_ could block the active sites on the photocatalyst’s surface. In turn, Li et al. [45] noticed that excess monosaccharide (glucose) leads to carbon overload on the photocatalyst’s surface. Consequently, the photocatalyst’s surface was covered by a thick layer of carbon, which hindered light absorption and reduced photoactivity [44].

Microorganisms are susceptible to increased levels of reactive oxygen species (ROS). Therefore, they have a mechanism consisting of various enzymatic antioxidants that neutralize the oxidative stress caused by the ROSs. Catalase (CAT) and superoxide dismutase (SOD) are two commonly investigated antioxidant intracellular enzymes that play essential roles in cell protection against reactive oxygen species [46,47]. This part of the work aimed to examine whether sucrose-modified TiO_2_ affects bacterial enzyme activity. Firstly, the blank experiments involving only bacteria in saline solution were carried out. No changes in bacterial CAT and SOD activity were observed in dark conditions and under ASL irradiation (Figure 9).

Under UV-A irradiation, the activity of CAT and SOD presented a slightly increasing trend and then fell back (Figure 9). However, compared to UV-A and the photocatalysts’ combined effect, the observed changes in enzymes activity were low and did not exceed 7% (Figure 9). As reported by Hoerter et al. [48], UV-A light influences the activity of oxidative defense enzymes and induces cytotoxicity dependent on the radiation intensity and dose distribution, not just the total energy dose.

As presented in Figure 10, the tested photocatalysts did not influence CAT and SOD activity during the experiment performed under dark conditions.

The activity levels of both CAT and SOD secreted by *E. coli* and *S. epidermidis* in experiments performed under UV-A and ASL irradiation are shown in Figure 11 and Figure 12, respectively.

Exposure to the C/TiO_2_ stimulated an antioxidative response and triggered CAT and SOD enzyme secretion. The best evidence is increased enzyme activity in the first 30 min of the photocatalytic process. The highest increases in CAT and SOD activity were observed for TiO_2_-S-1%-100 under UV-A conditions. After 30 min of irradiation, increases of 38% CAT and 26% SOD were noticed (for *E. coli*). In the case of *S. epidermidis,* the increases were 35% and 27% for CAT and SOD, respectively (Figure 11 and Figure 12). However, after 60 min, decreases in CAT and SOD activity were detected. Overaccumulation of ROS generated during the photocatalytic process with a saccharose-containing TiO_2_ overwhelmed the antioxidant capacity of SOD and CAT. After 60 min, the inhibitory rates of CAT were 22% for *E. coli* and 20% for *S. epidermidis*, while for SOD the rates were 35% and 30%, respectively. A similar fluctuation in antioxidant enzyme activity was reported in our previous study using APTES-modified TiO_2_ [49]. The deactivation kinetics of *Escherichia coli* and superoxide dismutase activity in photoreaction with titanium dioxide particles were examined by Koizumi et al. [49]. The authors suggested that the extremely short lifetime of exogenous oxidative radical species generated during photoreaction induces secondary oxidative stress inside the bacterial cells and causes a reduction in the intracellular superoxide dismutase activity, leading to bacteria death [50]. According to Zhang et al. [51], is possible that TiO_2_ nanoparticles bind to catalase via electrostatic or hydrogen bonding forces and induce changes in secondary and tertiary enzyme structures.

The TiO_2_-S-1%-100 photocatalyst had the most significant antibacterial activity against *E. coli* and *S. epidermidis* and exhibited good stability and reusability. It was also proved that the same disinfection efficiency was obtained after a few cycles. As is presented in Figure 13, TiO_2_-S-1%-100 remained effective and reusable after four continuous cycles of photocatalytic disinfection.

After the fifth cycle, a slightly slower bacteria inactivation was observed. Following five cycles, the decrease in activity might be attributed to bacteria cell deposition on the photocatalyst’s surface (Figure 9). However, it is worth mentioning that TiO_2_-S-1%-100 exhibited good stability even though it was washed or regenerated.

The photocatalytic inactivation of *E. coli* and *S. epidermidis* suspensions has also been modeled with four kinetic models (Chick–Watson model, modified Chick–Watson model, Hom model and modified Hom model) [52,53]. Based on the analysis of kinetic parameters, the modified Hom kinetic model showed the best fit to the results (a good R_2_ value, i.e., 0.999). The correlation coefficients (R_2_) and kinetic constants calculated from the kinetic analyses of four models for both bacteria under UV-A and ASL light are presented in the Appendix A. According to the Hom kinetic model, the kinetics of the photocatalytic death of *E. coli* and *S. epidermidis* are presented in Figure 14.

In agreement with the literature, many researchers have used the Hom model because it includes the lag phase bacteria in the description of the disinfection kinetics [52,53]. The modified Hom model is also the best empirical model representing the photocatalytic disinfection kinetics [52,53]. Therefore, it is assumed that only a modified Hom model can describe an initial delay resulting from the time required for the damage accumulation, a log-linear disinfection region and at the tail region. As shown in Figure 14, two regions are visible in the plot: a smooth decay at the beginning of the reaction, often called the “shoulder”, and a log-linear inactivation region that covers most of the reaction. The “shoulder” is related to an initial induction period where the production of radicals takes place (it lasts until the level of produced radicals becomes harmful to the bacteria) [53]. On the other hand, the least visible is the tail region representing a microbial subpopulation resistant to the disinfection.

However, only a few research articles are available on the different aspects of the monosaccharide (glucose)-modified titania and its photocatalytic properties and applications for water and air purification [23,24,54,55]. This study presents a novel synthetic route using the most common disaccharide–saccharose compounds as precursors for obtaining carbon-doped titania, which has been identified as an excellent way to overcome several of the challenges relating to the solar disinfection process.

## 3. Materials and Methods

### 3.1. Preparation of Saccharose-Modified Photocatalysts and Their Characterization

The same crude material (titania) and the same method were applied as described previously [16,17]. Due to an increase in annealing temperature resulting in decreased carbon content, titanium dioxide was treated with saccharose at an annealing temperature of 100 °C [16]. Three carbon-modified photocatalysts were obtained: TiO_2_-S-1%-100 (1% solution of saccharose), TiO_2_-S-5%-100 (5% solution of saccharose), and TiO_2_-S-10%-100 (10% solution of saccharose). The carbon source saccharose (C_12_H_22_O_11_) was purchased from Firma (Chempur^®^, Piekary Śląskie, Poland).

Fourier transform–infrared diffuse reflectance spectra (FTIR/DRS) of photocatalysts were recorded using an FTIR 4200 spectrometer (Jasco International Co. Ltd., Tokyo, Japan) equipped with a DR accessory from PIKE Technologies (Madison, WI, USA). The Brunauer–Emmett–Teller (BET) surface area values of the tested photocatalysts were measured using a low-temperature N_2_ adsorption–desorption method with a Quadrasorb SI analyzer (Anton Paar GmbH, Graz, Austria, Germany, previously Quantachrome Instruments, Boynton Beach, FL, USA). The total pore volume (V_total_) was determined based on the adsorbed N_2_ after finishing pore condensation at a relative pressure p/p_0_ = 0.99. Using the measured isotherm adsorption branches, the micropores volume (V_micro_) was defined using the Dubinin–Radushkevich equation. The mesopore volume (V_meso_) was calculated from the difference between V_total_ and V_micro_. The crystalline phase and crystal structure of obtained photocatalysts were analyzed via X-ray diffraction analyses (XRD) carried out with a PANalytical Empyrean X-ray diffractometer (Malvern PANalytical B.V., Almelo, the Netherlands) equipped with Cu Kα radiation (λ = 0.154056 nm). The crystallite size was determined using Scherrer’s equation [29]. The elemental contents of carbon in tested photocatalysts were determined using a CN 628 elemental analyzer (LECO Corporation, St. Joseph, MI, USA). The UV–Vis/DR spectra were recorded in the range of 250–800 nm using a V-650 UV–Vis spectrophotometer (JASCO International Co. Ltd., Tokyo, Japan) equipped with an integrating sphere accessory for studying DR spectra. The standard sample BaSO_4_ (purity 98%, Avantor Performance Materials Poland S.A., Gliwice, Poland) was used. The values of band gap energies (Eg) were calculated from the diffuse reflectance data by plotting the Kubelka–Munk function versus hυ.

To determine hydroxyl radical formation in the presence of C/TiO_2_ photocatalysts, the fluorescence technique using terephthalic acid (Acros Organics B.V.B.A, Gell. Belgium) was applied. For all tests, 0.01 g of the photocatalyst was suspended in 100 mL of the solution of terephthalic acid with an initial concentration of 0.083 g/L. Next, the suspension was exposed continuously to UV-A or ASL irradiation for 90 min. Sampling was performed every 10 min. Finally, the suspension (after filtration through a 0.45 μm membrane filter) was analyzed on a Hitachi F-2500 fluorescence spectrophotometer (Hitachi Group, Tokyo, Japan). The product of terephthalic acid hydroxylation, 2-hydroxyterephthalic acid (2-HTA), was detected as an emission peak at the maximum wavelength of 420 nm, with the excitation wavelength of 314 nm. For comparison, commercially available KRONOClean 7000 (Kronos International, Inc., Dallas, TX, USA) was used as a reference carbon-modified photocatalyst.

### 3.2. Microbiological Analysis

The antibacterial properties of the photocatalysts toward the Gram-negative *Escherichia coli* K12 (ATCC 29425) and Gram-positive *Staphylococcus epidermidis* (ATCC 49461) were determined using the method described previously [16]. The photocatalytic experiments were conducted under UV-A and artificial solar light (ASL). UV-A irradiation was provided by four Phillips Hapro Summer Glow bulbs with the power of 20 W each (Royal Phillips, Amsterdam, The Netherlands) and radiation intensity levels of 28.3 W/m^2^ in the spectral range of 300 to 2800 nm and 39.1 W/m^2^ in the spectral range of 280 to 380 nm. A 300 W light bulb (OSRAM Ultra Vitalux, OSRAM GmbH, Munich, Germany) with radiation intensity levels of 9.0 W/m^2^ in the spectral range of 300 to 2800 nm and 258.1 W/m^2^ in the spectral range of 280 to 380 nm was used as an ASL light source. The emission spectra measured using an Ocean Optics USB 4000 spectrometer (Ocean Optics Inc. Dunedin, FL, USA) are presented in the Appendix A.

A spectrophotometric method was used to determine superoxide dismutase (SOD) and catalase (CAT) activity levels. Assays were performed according to the method described previously [49]. The *E. coli* and *S. epidermidis* inactivation kinetics were evaluated with experimental data and four empirical models, namely the Chick–Watson model, modified Chick–Watson model, Hom model and modified Hom model, according to equations presented by Marugán et al. [53] and Cho et al. [56]. Analysis of the obtained results was performed using Statistica 13.3. The obtained R_2_ values from the kinetic analyses are presented in the Appendix A. In addition, the Appendix A present the kinetic constants for *E. coli* and *S. epidermidis* (Appendix A).

## 4. Conclusions

The green synthesis for the C-doped TiO_2_ using the intermediate product, taken directly from the production line of titanium(IV) oxide and saccharose (sucrose) solutions as the precursors for Ti and carbon sources at a low annealing temperature (100 °C), has been described. It was proven that this approach obtained photocatalysts with good surface characteristics and antibacterial activity under both UV-A and artificial solar light irradiation. Owing to the high bacterial death rate, bacterial oxidative enzyme inhibition and non-toxicity in dark conditions, saccharose-modified titanium dioxide can be applied for water disinfection. The antibacterial activity of C/TiO_2_ photocatalysts was strongly related to the presence of carbon in the structure of the material. TiO_2_-S-1%-100 containing 0.53 wt.% of carbon possessed the highest antibacterial activity against Gram-negative and Gram-positive model microorganisms presented in water. Moreover, it was proven that using widely available sucrose as the carbon precursor and a simple method without assembly and filling steps for fabrication of C-doped TiO_2_ active under artificial solar light (ASL) is possible. Hance, saccharose, as with D-glucose- and D-fructose-modified titania, can be implemented in UV-A or solar irradiation systems as an effective, inexpensive and safe photocatalytic disinfectant for industrial uses.

## Figures and Tables

**Figure 1 ijms-23-04719-f001:**
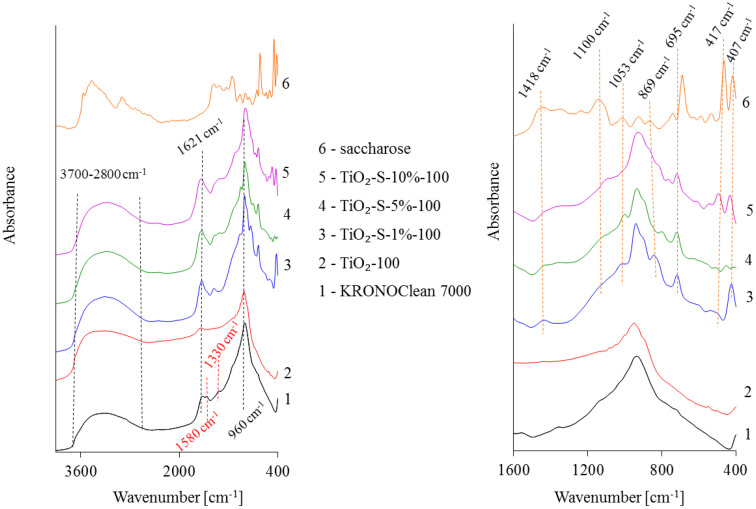
FT−IR/DR spectra of tested samples.

**Figure 2 ijms-23-04719-f002:**
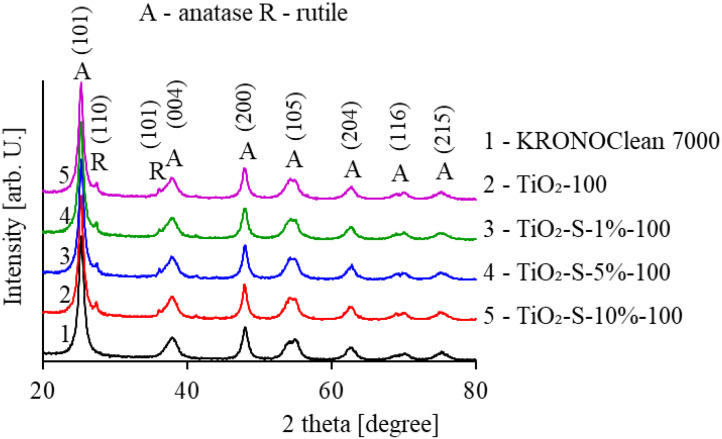
XRD patterns of the reference KRONOClean 7000, starting TiO_2_-100 and carbon-modified photocatalysts.

**Figure 3 ijms-23-04719-f003:**
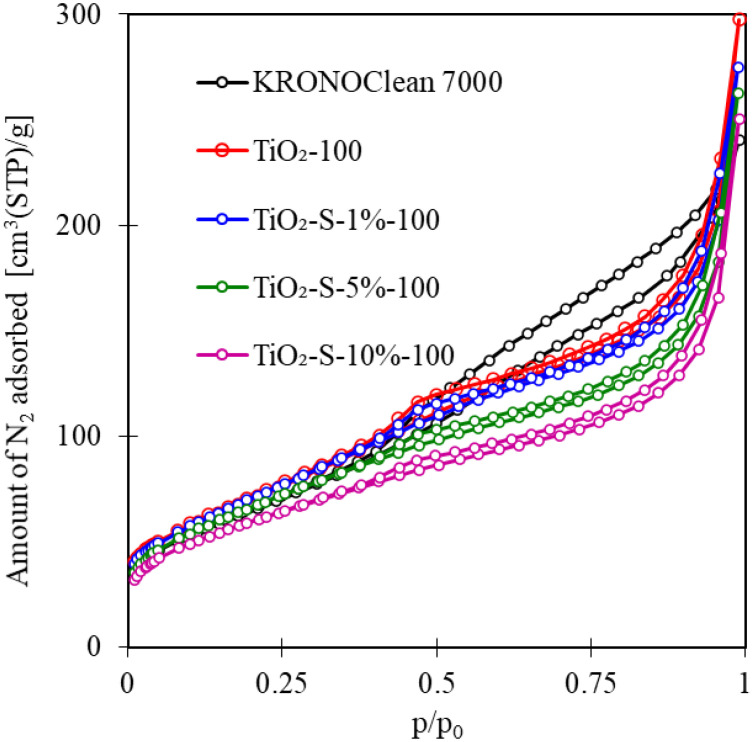
N_2_ adsorption–desorption isotherms for tested photocatalysts.

**Figure 4 ijms-23-04719-f004:**
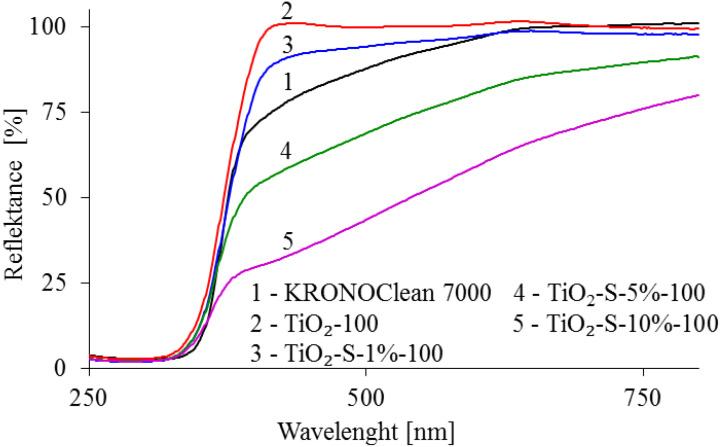
UV–Vis/DR spectra of tested samples.

**Figure 5 ijms-23-04719-f005:**
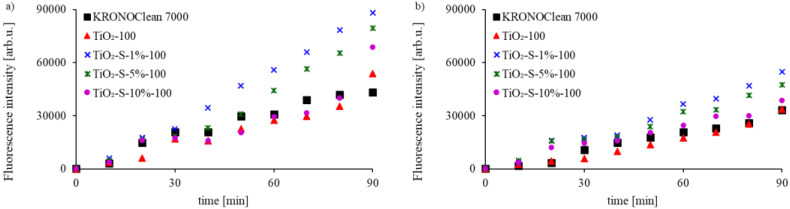
The fluorescence spectra of 2-hydroxyterephthalic acid product of the reaction of terephthalic acid with ·OH radical taken after different reaction times (**a**) under UV-A irradiation and (**b**) ASL irradiation.

**Figure 6 ijms-23-04719-f006:**
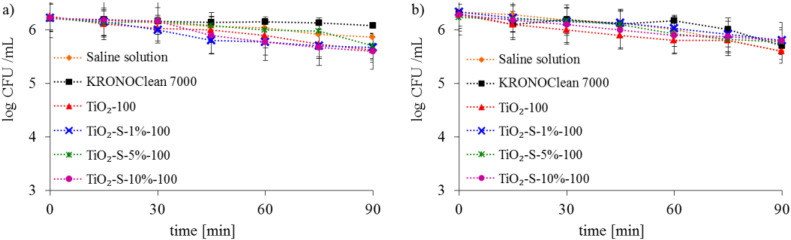
Effects of sucrose-modified titania on inactivation of bacteria in dark conditions: (**a**) Gram-negative *E. coli*; (**b**) Gram-positive *S. epidermidis*.

**Figure 7 ijms-23-04719-f007:**
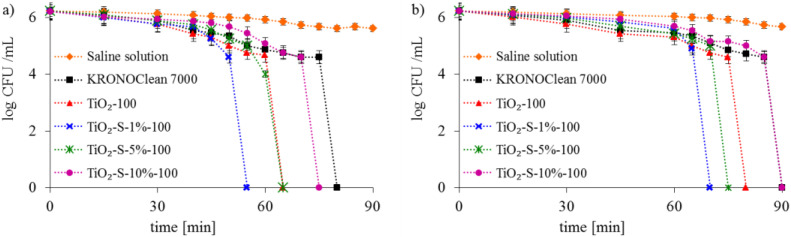
Inactivation of Gram-negative *E. coli* bacteria in the photocatalyst suspensions: (**a**) under UV-A; (**b**) under ASL irradiation.

**Figure 8 ijms-23-04719-f008:**
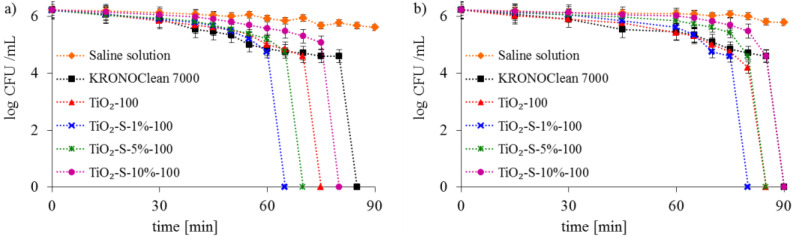
Inactivation of Gram-positive *S. epidermidis* bacteria in the photocatalyst suspensions: (**a**) under UV-A; (**b**) under ASL irradiation.

**Figure 9 ijms-23-04719-f009:**
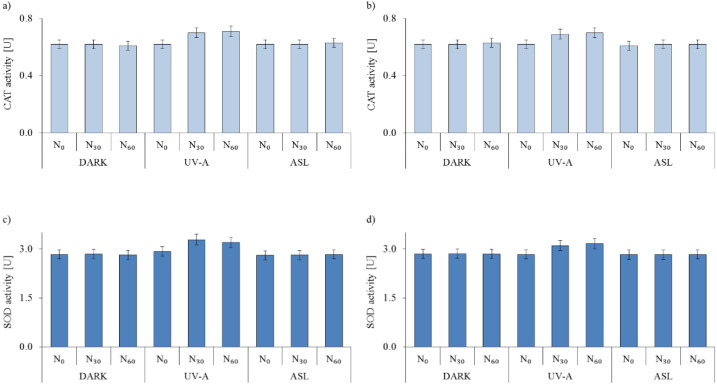
Activity of catalase (CAT) and superoxide dismutase (SOD) secreted by *E. coli* and *S. epidermidis* in dark conditions under UV-A or ASL: (**a**) CAT of *E. coli*; (**b**) CAT of *S. epidermidis*; (**c**) SOD of *E. coli*; (**d**) SOD of *S. epidermidis*.

**Figure 10 ijms-23-04719-f010:**
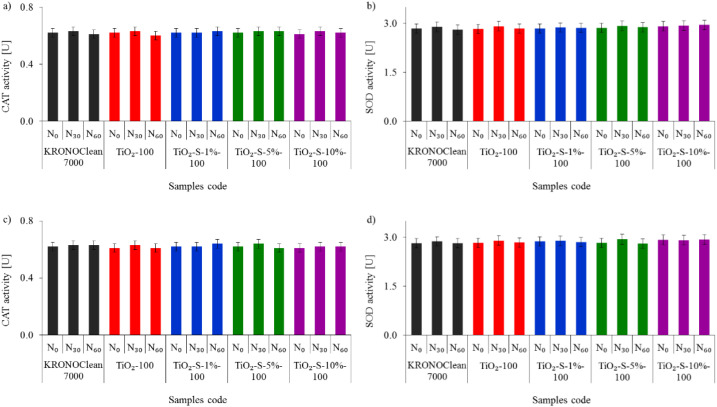
Influence of tested photocatalysts on the activity levels of bacterial catalase (CAT) and superoxide dismutase (SOD) in dark conditions: (**a**) CAT of *E. coli*; (**b**) SOD of *E. coli*; (**c**) CAT of *S. epidermidis*; (**d**) SOD of *S. epidermidis*.

**Figure 11 ijms-23-04719-f011:**
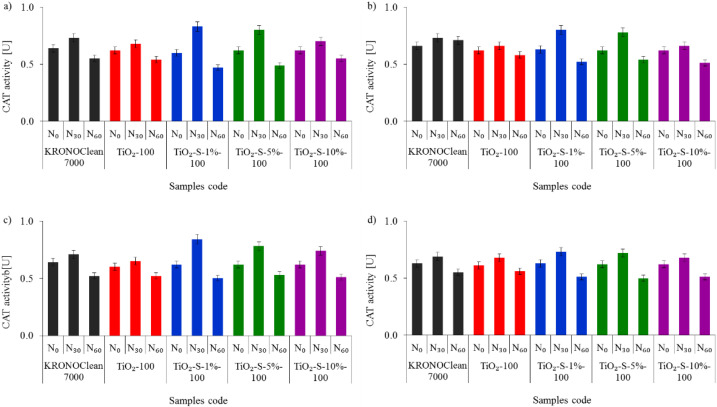
Influence of tested photocatalysts on the activity of bacterial catalase (CAT): (**a**) *E. coli* under UV-A; (**b**) *E. coli* under ASL; (**c**) *S. epidermidis* under UV-A; (**d**) *S. epidermidis* under ASL.

**Figure 12 ijms-23-04719-f012:**
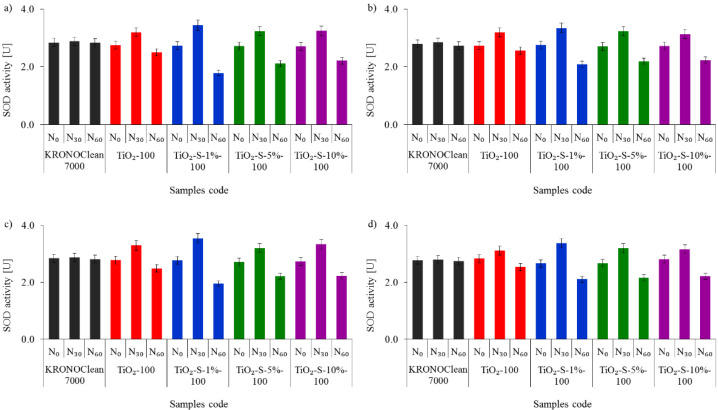
Influence of tested photocatalysts on the activity of bacterial catalase (SOD): (**a**) *E. coli* under UV-A; (**b**) *E. coli* under ASL; (**c**) *S. epidermidis* under UV-A; (**d**) *S. epidermidis* under ASL.

**Figure 13 ijms-23-04719-f013:**
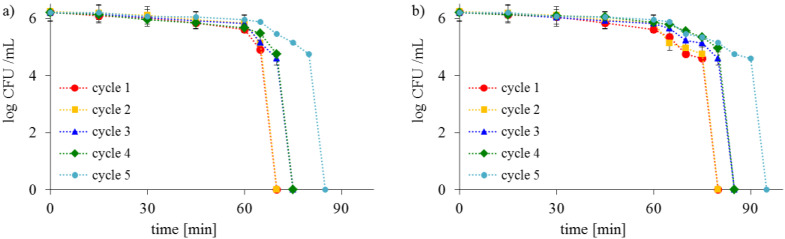
Inactivation of (**a**) *E. coli* and (**b**) *S. epidermidis* during the recycled photocatalytic process with TiO_2_-S-1%-100 under ASL irradiation.

**Figure 14 ijms-23-04719-f014:**
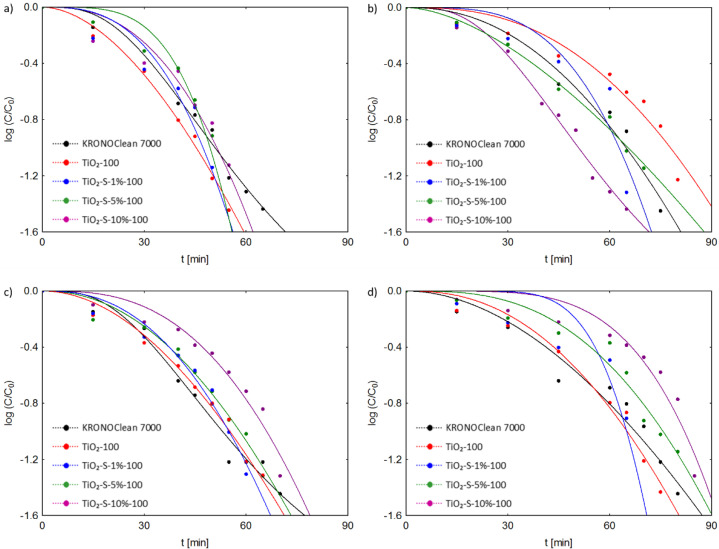
Fitting of Hom kinetic model to experimental data from the photocatalytic process conducted under UV-A or ASL irradiation: (**a**) *E. coli* (UV-A); (**b**) *E. coli* (ASL); (**c**) *S. epidermidis* (UV-A); (**d**) *S. epidermidis* (ASL).

**Table 1 ijms-23-04719-t001:** The phase compositions and average crystallite sizes of the reference KRONOClean 7000, starting TiO_2_-100 and carbon-modified photocatalysts.

Sample Code	TiO_2_ Crystalline Phase Participation [%]		Mean Crystallite Size [nm]	
	Anatase	Rutile	Anatase	Rutile
KRONOClean 7000	100	-	11.0	-
TiO_2_-100	98.1	1.9	12.0	52.8
TiO_2_-S-1%-100	97.9	2.1	11.7	42.3
TiO_2_-S-5%-100	97.8	2.2	11.7	42.3
TiO_2_-S-10%-100	97.8	2.2	11.6	42.3

**Table 2 ijms-23-04719-t002:** Structural parameters, carbon contents and Eg values of the studied photocatalysts.

Sample Code	S_BET_ [m^2^/g]	V_total(0.99)_ [cm^3^/g]	V_micro(DR)_ [cm^3^/g]	V_mezo_ [cm^3^/g]	Carbon Content (wt.%)	Eg [eV]
KRONOClean 7000	242	0.37	0.09	0.28	0.96	3.24
TiO_2_-100	266	0.46	0.09	0.37	-	3.25
TiO_2_-S-1%-100	264	0.43	0.09	0.34	0.53	3.21
TiO_2_-S-5%-100	261	0.45	0.08	0.37	2.42	3.20
TiO_2_-S-10%-100	217	0.39	0.04	0.34	4.40	2.92

## Data Availability

The data presented in this study are available on request from the corresponding author.

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
