# Peer review of "The Benefits of Using Saccharose for Photocatalytic Water Disinfection"

_ijms, 2022, doi:10.3390/ijms23094719_

Round 1
Reviewer 1 Report
In this manuscript by Paulina et al., sucrose modified TiO2 photocatalysts were prepared by hydrothermal method and tested for bacteria survivability and enzyme activity under UV-A and artificial solar light. The mythologies used in this manuscript are largely the same with previous studies, which limited the novelty of the present work. In addition, the reviewer also noticed some inconsistence and misinterpretation of the results in the discussions, which will be commented below:
- In the abstract, the authors stated that the obtained TiO2-1%-S-100 photocatalysts were green color, however, in page 6 line 162: “TiO2-100 and TiO2-S-1%-100 stayed white”, the descriptions were inconsistent.
- In page 6, the authors trying to explain the difference in photo-activity of sucrose, glucose, and fructose modified TiO2 by their different solubility in aqueous solutions. However, it is unclear what is the order of the solubility and why the solubility is related with the photo-activity.
- In Figure 7 and Figure 8, why the curves show a steady decrease at the beginning, but a sudden drop at the last point?
- In page 8 line 221, the author asserted that: “The bacteria inactivation was observed only in experiments conducted with sucrose modified TiO2 under UV-A and ASL irradiation. All photocatalysts presented better anti-bacterial properties than starting TiO2 .”. This statement is obviously not true, as seen in Figure 7 and Figure 8, starting TiO2 also exhibit activity for bacteria inactivation, and even shows better activity than TiO2-S-10%-100.
- The authors shown that the lowest concentration of sucrose yields the best photo-activity in the current study and concluded that monosaccharides or disaccharides at low concentrations are good carbon sources for synthesizing C/TiO2 photocatalyst. However, most importantly, what is the optimal concentration was not shown. Will further decrease the sucrose concentration increase the activity?
- The photocatalysts start to dramatically decrease activity after 4 cycles. With this limited stability, the reviewer do not think the photocatalyst exhibited good stability and reusability, as the author claimed.
- More discussion on the kinetic studies are needed. From Figure 14, it looks like all photocatalysts show a similar kinetic, and it is hard to say which is better than the other.
Author Response
Dear Reviewer 1,
Thank you for many valuable comments on our manuscript, and we would like to take this opportunity to express our great appreciation for him/her and the comments. Followings are our responses to the reviewer’s questions and comments and the changes made in the revised version (marked with red color).
- In the abstract, the authors stated that the obtained TiO2-1%-S-100 photocatalysts were green color, however, in page 6 line 162: “TiO2-100 and TiO2-S-1%-100 stayed white”, the descriptions were inconsistent.
We want to thank the reviewer for this valuable comment. But, of course, this is likely just a case of inattentiveness, and we will correct it in the text.
“Obtained TiO2-1%-S-100 photocatalysts were capable of total E. coli and S. epidermidis inactivation under ASL irradiation with less than 1 h.”
- In page 6, the authors trying to explain the difference in photo-activity of sucrose, glucose, and fructose modified TiO2by their different solubility in aqueous solutions. However, it is unclear what is the order of the solubility and why the solubility is related with the photo-activity.
The answer to why we tried to explain the difference in photoactivity of sucrose, glucose, and fructose modified TiO2 by their different solubility is not straightforward. In the authors' opinion, it is possible that the solubility of sugars can influence on coverage rate of titania. It is well known that enhancing the solubility of dopants can improve the stability of desired defects [1]. The effect of dopant solubility on the photoactivity of titania was discussed by Wen-Fan Chen et al. for Ce/Cr [2]. As showed by Gonell F. et al. [3], the solubility of the metallic cation increases, and the availability of copper to be incorporated into the nascent TiO2 lattice. The solubilities from most soluble to least soluble were fructose, sucrose and glucose. The values also significantly depend on temperature.
[1] Zhang, J., Tse, K., Wong, M. et al. A brief review of co-doping. Front. Phys. 11, 117405 (2016). https://doi.org/10.1007/s11467-016-0577-2.
[2] Chen, W. F., Mofarah, S. S., Hanaor, D. A. H., Koshy, P., Chen, H. K., Jiang, Y., Sorrell, C. C. (2018). Enhancement of Ce/Cr codopant solubility and chemical homogeneity in TiO2 nanoparticles through sol–gel versus Pechini syntheses. Inorg. Chem. 57(12), 7279-7289. https://doi.org/10.1021/acs.inorgchem.8b00926.
[3] Gonell, F., Puga, A. V., Julian-Lopez, B., Garcia, H., Corma, A. (2016). Copper-doped titania photocatalysts for simultaneous reduction of CO2 and production of H2 from aqueous sulfide. Appl. Catal. B: Environ. 180, 263-270. https://doi.org/10.1016/j.apcatb.2015.06.019.
- In Figure 7 and Figure 8, why the curves show a steady decrease at the beginning, but a sudden drop at the last point?
Thank you for your valuable suggestion. The changes in bacteria inactivation curves probably are ascribed to the two-step mechanism of bacteria destruction in a photocatalyst. This mechanism has been described, i.e. by Desai and Kowshik (2009). The authors observed that the bacteria inactivation had occurred in two distinct phases. In the first stage, bacteria could trigger self-repair and self-defence mechanisms and are resistant to ·OH radical’s attack. At this stage, the cells did not lose their viability. Therefore, during the first 60 min of the photocatalytic process, no significant changes in the bacteria number were observed. Next, the photocatalytic process made the cell membranes and walls permeable, and intracellular cytoplasmic components leaked from the cells. Free TiO2 nanoparticles might also diffuse into the cell and cause damage inside cells. As a consequence of a direct attack on intracellular components, rapid bacteria inactivation was observed. Such changes in bacteria inactivation rate are also observed in other works, e.g Rincón and Pulgarin (2003), and Wanag et al. (2016).
[4] V.S. Desai and M. Kowshik, Antimicrobial activity of titanium dioxide nanoparticles synthesized by sol-gel technique. Res. J. Microbiol. 4(3) (2009) 97-103. https://doi.org/10.3923/jm.2009.97.103.
[5] A.G. Rincón, C. Pulgarin, C. (2003). Photocatalytical inactivation of E. coli: effect of (continuous–intermittent) light intensity and of (suspended–fixed) TiO2 concentration. Appl. Catal. B: Environ. 44(3) (2003) 263-284. https://doi.org/10.1016/S0926-3373(03)00076-6.
[6] A. Wanag, P. Rokicka, E. Kusiak-Nejman, A. Markowska-Szczupak, A. Morawski, TiO2/glucose nanomaterials with enhanced antibacterial properties. Mater. Lett. 185 (2016) 264-267. https://doi.org/10.1016/j.matlet.2016.08.133.
- In page 8 line 221, the author asserted that: “The bacteria inactivation was observed only in experiments conducted with sucrose modified TiO2 under UV-A and ASL irradiation. All photocatalysts presented better anti-bacterial properties than starting TiO2”. This statement is obviously not true, as seen in Figure 7 and Figure 8, starting TiO2also exhibit activity for bacteria inactivation, and even shows better activity than TiO2-S-10%-100.
We thank the referee for this comment. We have improved this section. The bacteria inactivation was observed in experiments conducted with starting TiO2 and sucrose modified TiO2 under UV-A and ASL irradiation. However, only TiO2-S-1%-100 photocatalysts presented better antibacterial properties than the starting TiO2 under UV-A, whereas TiO2-S-1%-100 and TiO2-S-5%-100 under ASL.
Appropriate corrections have been made. “The bacteria inactivation was observed in experiments conducted with commercial KRONOClean 7000, starting TiO2 and sucrose modified TiO2 under UV-A and ASL irradiation. The strongest antibacterial activity was presented for photocatalyst obtained by modification with 1 wt.% of saccharose solution (TiO2-S-1%-100). Total E. coli inactivation was achieved after 55 min and 65 min photocatalytic process under UV-A and ASL irradiation, respectively (Figure 7). Gram-positive Staphylococcus epidermidis was more invulnerable to photocatalytic disinfection. Total bacteria inactivation was obtained in the same conditions after 65 under UV-A and after 80 min under ASL. TiO2-S-1%-100 photocatalysts presented better antibacterial properties than starting TiO2 in all experiments. It is worth mentioning that commercial KRONOClean 7000 cause E. coli and S. epidermidis in-activation in 80 and 85 min under UV-A or 90 minutes under ASL irradiation (Figures 7, 8).”
- The authors shown that the lowest concentration of sucrose yields the best photo-activity in the current study and concluded that monosaccharides or disaccharides at low concentrations are good carbon sources for synthesizing C/TiO2photocatalyst. However, most importantly, what is the optimal concentration was not shown. Will further decrease the sucrose concentration increase the activity?
Thank you for your valuable comments and suggestions on the purpose of our studies. We did not check if a further decrease of sucrose concentration increases the photoactivity. We also agree that examining lower concentrations of sucrose may help better understand this process. Unfortunately, we cannot perform these experiments due to limited review response time, but we will include them in future research.
- The photocatalysts start to dramatically decrease activity after 4 cycles. With this limited stability, the reviewer do not think the photocatalyst exhibited good stability and reusability, as the author claimed.
The stability of the photocatalytic activity is one of the most critical aspects of an industrial process; increasing the catalyst lifetime and reusing it simplifies the processes and lower its overall cost. We assumed that photocatalyst exhibited good stability due to some authors reported even a 30% loss after re-use (Ramos D.R., Iazykov M., Fernandez M.I., Santaballa J.A., Canle M., Mechanical Stability Is Key for Large-Scale Implementation of Photocatalytic Surface-Attached Film Technologies in Water Treatment, Frontiers in Chemical Engineering,3). However, more work is needed on photocatalyst longevity under operational and real conditions.
- More discussion on the kinetic studies are needed. From Figure 14, it looks like all photocatalysts show a similar kinetic, and it is hard to say which is better than the other.
Thank you for your valuable suggestion. Appropriate corrections have been made.
“The photocatalytic inactivation of E. coli and S. epidermidis suspensions have also been modelled with four kinetic models (Chick-Watson model, modified Chick-Watson model, Hom model and modified Hom model) [52, 53]. Based on the analysis of kinetic parameters, the modified Hom kinetic model showed the best fit to the results (a good R2 value, i.e. 0.999). The correlation coefficients (R2) and kinetic constants calculated from the kinetic analyses of four models for both bacteria under UV-A and ASL light were presented in supplementary materials (Table SM 1 and SM 2). According to Hom kinetic model, the kinetics of photocatalytic death of E. coli and S. epidermidis were presented in Figure 14. In agreement with literature, many researchers have used Hom model because it included the lag phase bacteria in the description of disinfection kinetics [52, 53]. The modified Hom model is also the best empirical model representing the photocatalytic disinfection kinetics [52, 53]. Therefore, it is assumed that only a modified Hom model can describe an initial delay resulting from the time required for the damage accumulation, a log-linear disinfection region and at the tail region. As shown in Figure 14 two regions were visible in the plot: a smooth decay at the beginning of the reaction, often called “shoulder” and a log-linear inactivation region that covers most of the reaction. The “shoulder” is related to an initial induction period where the production of radicals takes place (it lasts until the level of produced radicals becomes harmful to the bacteria) [53]. On the other hand, the least visible is the tail region representing a microbial subpopulation resistant to the disinfection.”

Reviewer 2 Report
This manuscript mainly discusses the sucrose-modified titanium dioxide photocatalysts and their disinfection efficiency on E. coli and S. epidermidis system under UV-A and ASL. the authors did a good job of experimental design, data analysis, and the results presenting. However, some minor revisions are still needed before accepting-
- for figure #3, both a and b figures should include all five tested photocatalysts to show the main difference
- since the main target of this manuscript is talking about the sucrose-modified titanium dioxide photocatalyst, it might be better to modify the title of the manuscript, using a specific work instead of "sugars"
- since the title is about the benefits of the photocatalytic they generated in the water disinfection, it could be nice if the authors could emphasize the benefits of these catalysts compared to the other in the manuscript
- If the manuscript can use some more recent references that have been published in the last three years, it would be nice.
Author Response
Dear Reviewer 2,
Thank you for many valuable comments on our manuscript, and we would like to take this opportunity to express our great appreciation for him/her and the comments. Our responses to the reviewer’s questions and comments and the changes made in the revised version (marked with blue color).
- For figure #3, both a and b figures should include all five tested photocatalysts to show the main difference
Thank you for your valuable suggestion. Appropriate corrections have been made.
- Since the main target of this manuscript is talking about the sucrose-modified titanium dioxide photocatalyst, it might be better to modify the title of the manuscript, using a specific work instead of "sugars"
Thank you for your helpful comment and suggestion on the title. We decided modified the title according to your suggestion.
“The benefits of using saccharose for photocatalytic water disinfection.”
3.Since the title is about the benefits of the photocatalytic they generated in the water disinfection, it could be nice if the authors could emphasize the benefits of these catalysts compared to the other in the manuscript
We want to thank the reviewer for the positive mark, but in the last sentence of the Conclusions section, we summarized that “using wildly available sucrose as carbon precursor and simple method without assembly and filling steps were introduced in this work for the fabrication of C-doped TiO2. Hance, saccharose, just as D-glucose and D-fructose, modified titania, can be implemented in the UV-A or ASL irradiation systems as an effective, inexpensive and safe photocatalytic disinfectant for industrial uses”.
- If the manuscript can use some more recent references that have been published in the last three years, it would be nice.
Thank you for your valuable suggestion. According to the Reviewer’s suggestion, we cited some more recent references.
[2] Karim, M.R., Khan, M.H.R.B., Akash, M.A.S.A., Shams, S. Effectiveness of solar disinfection for household water treatment: an experimental and modeling study. J. Water Sanit. Hyg. Dev. 2021, 11(3), 374-385.
[3] Chaúque, B.J.M., Rott, M. B. Solar disinfection (SODIS) technologies as alternative for large-scale public drinking water supply: Advances and challenges. Chemosphere, 2021, 281, 130754.
[4] Duan, X., Zhou, X., Wang, R., Wang, S., Ren, N.Q., Ho, S.H. (2021). Advanced oxidation processes for water disinfection: Features, mechanisms and prospects. Chem. Eng. J. 2021, 409, 128207.
[5] Mahy, J.G., Wolfs, C., Vreuls, C., Drot, S., Dircks, S., Boergers, A., Tuerk, J., Hermans, S., Lambert, S.D. Advanced oxidation processes for waste water treatment: from laboratory-scale model water to on-site real waste water. Environ. Technol. 2021, 42(25), 3974-3986.
[6] Tan, L.L., Wong, V.L., Phang, S.J. Recent advances on TiO2 photocatalysis for wastewater degradation: fundamentals, commercial TiO2 materials, and photocatalytic reactors. Handbook of Nanotechnology Applications, 2021, 25-65.
[7] Magaña-López, R., Zaragoza-Sánchez, P.I., Jiménez-Cisneros, B.E., Chávez-Mejía, A.C. The use of TiO2 as a disinfectant in water sanitation applications. Water, 2021, 13(12), 1641.
[9] Silva, T.F., Peri, P., Fajardo, A.S., Paulista, L.O., Soares, P.A., Martínez-Huitle, C.A., Vilar, V.J. Solar-driven heterogeneous photocatalysis using a static mixer as TiO2-P25 support: impact of reflector optics and material. Chem. Eng. J. 2022, 134831.
[10] Alkorbi, A.S., Javed, H.M.A., Hussain, S., Latif, S., Mahr, M.S., Mustafa, M.S., Alsaiari, ., Alhemiary, N. A. Solar light-driven photocatalytic degradation of methyl blue by carbon-doped TiO2 nanoparticles. Opt. Mater. 2022, 127, 112259.
[11] Hua, L., Yin, Z., Cao, S. Recent advances in synthesis and applications of carbon-doped TiO2 nanomaterials. Catalysts, 2020, 10(12), 1431.
[12] Piątkowska, A., Janus, M., Szymański, K., Mozia, S. C-, N-and S-doped TiO2 photocatalysts: a review. Catalysts, 2021, 11(1), 144.

Round 2
Reviewer 1 Report
The authors have addressed most of my concerns, while as the authors stated in the response, more work in the future are needed to provide further understanding.